# Pollutants Concentration during the Construction and Operation Stages of a Long Tunnel: A Case Study of Lowari Tunnel, (Dir–Chitral), Khyber Pakhtunkhwa, Pakistan

**Jehanzeb Khan [1,2], Waqas Ahmed [1], Muhammad Yasir [1], Ihtisham Islam [1,3], Hammad Tariq Janjuhah [3,*] and George Kontakiotis [4,*]**

1 National Centre of Excellence in Geology, University of Peshawar, Peshawar 25120, Pakistan; jehanzebgeology@gmail.com (J.K.); waqas.nce@uop.edu.pk (W.A.); myasir6755@uop.edu.pk (M.Y.); ihtisham.islam@sbbu.edu.pk (I.I.)
2 Department of Geology, University of Malakand, Chakdara, Dir (L) 18800, Pakistan
3 Department of Geology, Shaheed Benazir Bhutto University Sheringal, Dir (U) 18000, Pakistan
4 Department of Historical Geology-Paleontology, Faculty of Geology and Geoenvironment, School of Earth Sciences, National and Kapodistrian University of Athens, Panepistimiopolis, Zografou, 15784 Athens, Greece
* Correspondence: hammad@sbbu.edu.pk (H.T.J.); gkontak@geol.uoa.gr (G.K.)

**Abstract:** Long tunnels with significant overburden, changeable geological conditions, a steep gradient, water infiltration, and heavy traffic flow are susceptible to environmental concerns during both construction and operation. The availability of fresh air and visibility is the most important necessity in excavation for tunnel workers inside the tunnel during the construction phase, as well as those crossing the tunnel during operation. Lowari Tunnel's tunnel air pollutants were researched. Carbon monoxide (CO), carbon dioxide ($CO_2$), oxygen ($O_2$), hydrogen sulfide ($H_2S$), sulfur dioxide ($SO_2$), nitrogen oxide (NO), ammonia ($NH_3$), nitrogen dioxide ($NO_2$), $PM_1$, $PM_{2.5}$, $PM_{10}$, air velocity, dust morphological and particle size distribution analysis are among the parameters under consideration. The findings provide evidence for the development of tunnel air quality.

**Keywords:** road tunnel; construction stage; operational stage; gases concentration; particulate matter; environmental monitoring and engineering

## 1. Introduction

During road tunnel construction and operation, pollutants, such as hazardous gases and particulate matter, may represent a substantial health risk [1]. During tunnel construction, these pollutants are produced by blasting, mucking, and concreting [2,3]. A higher concentration of hazardous gases is reported near the working face of the tunnel that gradually decreases with an increase in distance. During the operating stage, they are mostly generated by traffic flow [4–6]. The investigation of emissions is vital in preventing tunnel pollution because of their health consequences and source identification [7]. Two sorts of emissions exist: exhaust and non-exhaust. Exhaust emissions come from vehicle tailpipe, whereas non-exhaust emission comes from wear and tear (brakes, tires, and clutches), re-suspension, road surface dust, and abrasion. Exhaust emissions contribute to fine particulate matter, ($PM_{2.5}$ and smaller), while non-exhaust emissions contribute to the coarser particulate matter ($PM_{2.5}$–$PM_{10}$) [8]. In general, non-exhaust emissions are defined by trace elements (e.g., Cu, Zn, Ba, Sb, Mn) but organic markers have been used in certain instances [9]. Even in developed countries, current levels of non-exhaust emissions and future forecasts are on the rise. Lima et al. [10] observed that the link between traffic volume and non-exhaust emissions is non-linear. These emissions are affected by vehicles and the road conditions; low traffic volumes in bad conditions cause more pollution than high traffic volumes in good conditions. Reduced traffic paired with poor tunnel road conditions will also increase pollution inside the tunnel. The situation further deteriorates

when the traffic volume is high and tunnel road conditions are poor. In addition to $H_2S$, NO, and $SO_2$, other gases may trigger explosions in road tunnels [11]. Similarly, $H_2S$ concentrations exceeding 10 ppm may produce headaches, dizziness, nausea, burning eyes, sore throats, and other respiratory symptoms [12], while concentrations exceeding 500 ppm are deadly [13,14]. Particulate matter contributes significantly to cardiovascular disease, lung disease, and atherosclerosis [15,16]. Road traffic is the primary source of PM concentration and the principal contributor of ultrafine and fine nanoparticles of Ba, Zn, and Pb [17], posing a cancer risk to the exposed human population [18,19]. $PM_{2.5}$ reduces tunnel vision, which may result in fatal accidents [20].

In short tunnels ($\leq 1$ km), natural airflow is usually sufficient to flush out impurities [21]. However, in long tunnels, it is essential to provide fresh air throughout the construction and operational stages to maintain acceptable levels of harmful gases [22–28]. Lokhandwala and Gautam [29] demonstrated that rearranging traffic lights along with a ventilation system with a flow rate of 40,000 $m^3$ per hour substantially reduced pollution levels. In addition, pollution control vehicles and $NO_2$ absorption technologies may be used to boost airflow and decrease photochemical smog concentrations, which are influenced by traffic volume and vehicle fleet composition [4,30,31]. A minimum air velocity of 0.3 m/s is necessary for a clean tunnel environment. Nevertheless, external factors such as rain or snow may interrupt airflow inside the tunnel, thereby increasing the pollutants' concentration [32]. However, Hung-Lung and Yao-Sheng [33] showed that during rainy days, lower particle concentrations can be observed due to reduced particle re-entrainment and re-suspension in the tunnel. The tunnel's airflow is proportional to its cross-section [34]. During the construction stage of a tunnel, a larger-diameter duct is installed to reduce airflow resistance; otherwise, dust levels may exceed 2000 mg/$m^3$ [35]. In such settings, the health of employees and people passing through the tunnel may be at risk [36]. The air quality and wind direction outside the tunnel have a substantial effect on the air quality within the tunnel.

Recent global research has measured gaseous air pollutants and particle matter in road tunnels in Guanajuato, Mexico [37]; in Tehran, Iran [38]; and China [39]. Given the lack of attention to international environmental rules, the low quality of buildings, and the bad condition of vehicles, such studies are particularly essential in developing nations such as Pakistan. The strategic location and significance of the Lowari Tunnel necessitates the monitoring of pollutants (including gas concentrations and particulate matter), oxygen levels, and airflow velocity to determine the tunnel's long-term sustainability. This is, to the best of our knowledge, the first research to report the concentration of gaseous and particulate matter during the construction and operation of a tunnel in Pakistan. During the construction stage, gases such as CO, $CO_2$, $O_2$, $H_2S$, $SO_2$, and NO were monitored, while during the operational stage, CO, $CO_2$, $O_2$, $H_2S$, $SO_2$, NO, $O_3$, $NH_3$, $NO_2$, and particulate matter ($PM_{10}$, $PM_{2.5}$, and $PM_1$) were monitored. In addition, the morphology, elemental content, and particle size distribution of tunnel dust were also investigated.

## 2. Materials and Methods

Lowari Top is situated at a height of 3100 m and connects Chitral in the north to other regions of Pakistan in the south (Figure 1). Due to significant snowfall throughout the winter season, it remains closed. This terrain is characterized by snow avalanches, rock-falls, and landslides, making road building and maintenance challenging and expensive. Consequently, in 1960, a horseshoe-shaped rail tunnel with a length of about 8.5 km and a cross-section size of 52.3 $m^2$ was planned [40]. The construction started in 1975 and ended in 1977, 2 years later. After a lengthy delay, efforts resumed in 2006, and in 2009, a breakthrough was made. In 2012, the design of the rail tunnel was changed to a road tunnel with a cross-section size of 87.2 $m^2$. In 2017, the building of the modified road tunnel was finished, and it began operating. The south portal of the Lowari Tunnel is situated at 35°18′59.39″ N and 71°50′08.17″ E, while the north portal is located at 35°22′41.11″ N and 71°47′05.19″ E. This tunnel connects the districts of Dir and Chitral in Khyber Pakhtunkhwa,

Pakistan. The tunnel has a south-to-north slope of 1.8% and crosses many geological units, such as granite, granite biotite, metasediments, meta-igneous rocks, meta-volcanic, and shear zones. It has a maximum overburden of 1100 m and considerable water infiltration. In the summer, the average daily traffic volume is 350 vehicles and 2000 passengers, while in the winter, it drops to 200 vehicles and 1000 passengers. The vehicle traffic consists of automobiles (50%), trucks (40%), and four-wheel-drive vehicles (10%).

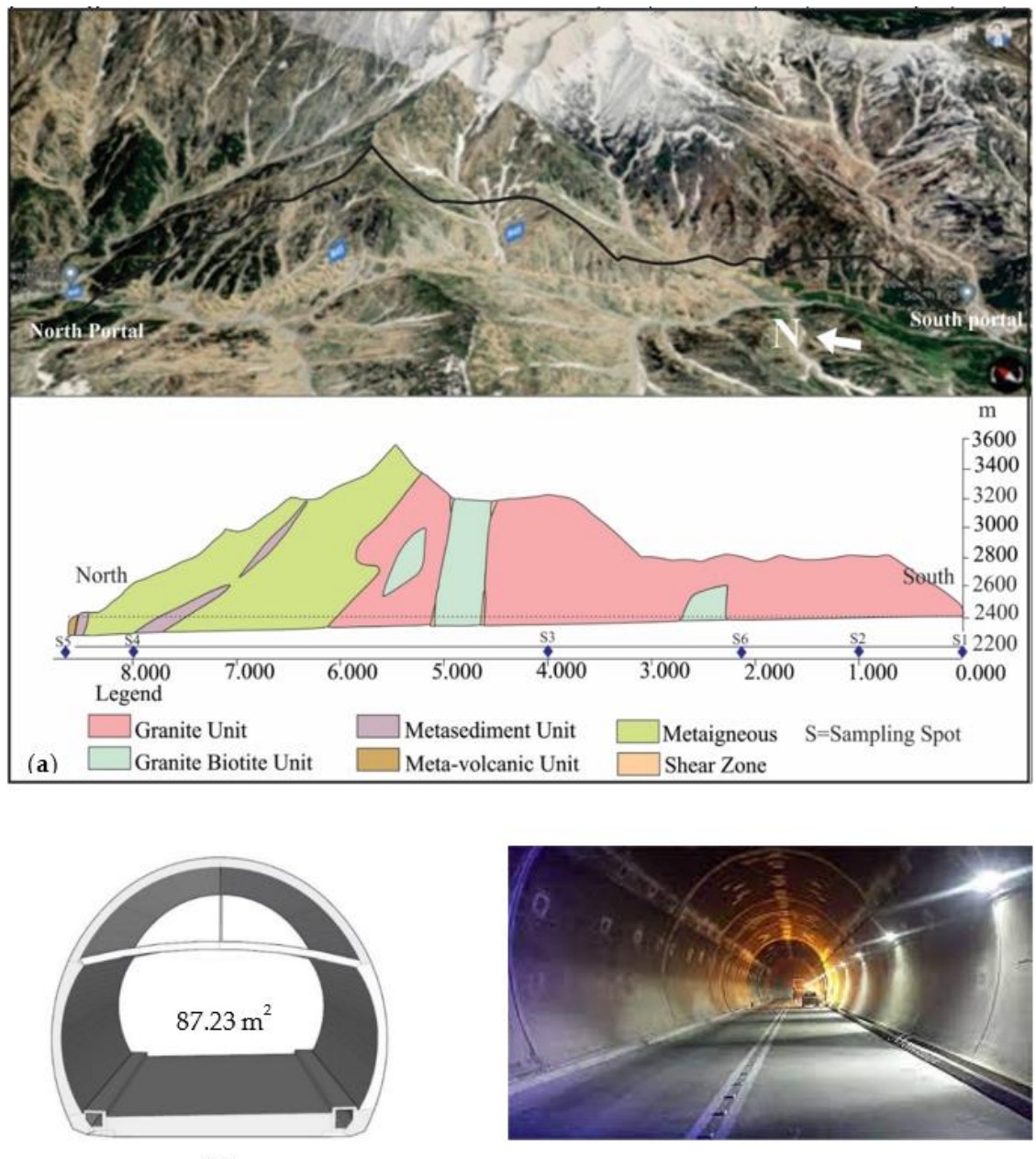

**Figure 1.** (**a**) Tunnel length, geological units and overburden (modified from Ur Rehman, et al. [40]. (**b**) Lowari Tunnel's profile. (**c**) Real-time image.

## 2.1. Construction Stage

In November 2008, December 2008, and January 2009, between Chainage 4 + 369.8 and 4 + 595, $H_2S$, $SO_2$, and NO concentrations were monitored for 30 days, while CO, $CO_2$, and $O_2$ concentrations were monitored for 32 days. The IR multi-gas monitor-900614 was used to measure CO, $CO_2$, and $O_2$ gases. The gases pass through an active filter

that absorbs infrared light. A reference filter allows infrared light to pass that does not interact with the target gas molecules. Two infrared beams of different wavelengths are transmitted into the measurement chamber, which houses the measuring and reference detectors. The lower intensity of a measurement beam correlates to the gas concentration. The RAE Systems V-Rae PGM-7840 multi-gas detector with an electrochemical sensor and monitors was utilized to measure $H_2S$, $SO_2$, and NO gases. Both portable multi-gas monitors were calibrated by the manufacturer and operated by RAE System Inc., San Jose, CA, USA (Document 008-4028, 2001). The six monitored gases were selected on the basis of the likelihood of their existence inside the tunnel throughout construction, enabling concentrations to be compared before and after blasting. Following tunnel blasting, the data were obtained within 250 m of the working face. For each day, monitoring of the gases was conducted with a monitoring duration of 1 h and a sampling frequency of 60 s, and the average values are reported. Through use of the Pro RAE-Suite Software, the collected readings were then compiled and analyzed. The monitoring data for gases were processed and studied to identify the concentration trends and compared with the Threshold Limits of Safety and Administration (2003).

### 2.2. Operational Stage

During the tunnel's operational stage on 30 June 2021, hazardous gases and tunnel dust were analyzed at four spots: Spot 1 (south portal), Spot 2 (Chainage 1 + 000), Spot 3 (Chainage 4 + 000), and Spot 4 (Chainage 8 + 000). In addition to these spots, dust from the tunnel roadside was also collected at Spot 5 (north portal). Each dust sample (about 1000 g) was collected with a bristle brush in a tightly sealed plastic bag. Through use of the the NOVA AMA instrument (600-2-3-4-5-7-10), hazardous gases such as $H_2S$, NO, CO, $CO_2$, $NH_3$, and $NO_2$ were monitored. The gas was pumped into the sample chamber, and the concentration of the gas was detected electro-optically by its absorption of a specific infrared (IR) wavelength. Similarly, $O_2$ was measured by a Stack gas analyzer (Int-Lancom-III) that uses electrochemical, infrared, and pellistor/catalytic sensors. $O_3$ was measured with an ozone monitor (C-30 ZX) that has a detection range of 0.02–0.14 ppm. The HAZ Dust Analyzer was used to monitor $PM_{10}$, $PM_{2.5}$, and $PM_1$ particulate matter. It works by scattering near-forward light from infrared radiation. The light is scattered as the airborne particles enter the infrared beam. The amount of light received by the photodetector is proportional to the concentration of the aerosol. These instruments were used in compliance with the instrumentation standards of Cornelis [41]. Each instrument was set up for 1 h of recording at a 1-s frequency, and the average values are reported.

The morphology and elemental concentration of the tunnel dust was studied by scanning electron microscopy (SEM) and an energy dispersive X-ray system (EDS) using a Jeol SEM vacuum chamber machine (JSM-IT-100). Microphotographs with magnifications of 65 to 250× and pixel sizes of 1280 × 960 were taken for each sample. The apparatus was run in a high vacuum with a 20 kV acceleration voltage. The EDS microanalysis was carried out with a capture duration of 60 s, a count rate of 2226 to 2315 counts per second, and a dead time of 3–4%. To determine the major elements (Si, Ca, Na, Mg, Fe, and K) and trace elements (Zn, Pd, Ni, Ag, Co, and Cd), atomic absorption spectroscopy (AAS) with Perkin Elmer-700 and UV/VIS spectrometer SP-400 machines was applied. For major elements, a 0.5 g sample was obtained and dissolved in the stock solution using multi-acid (HF, HClO4, and HCl) digestion. For trace elements, a 1 g sample was dissolved in aqua regia and then diluted in 50 mL of distilled water. The precision was calculated using the reference standards G2, AGVI, and W-2. The analysis achieved a 95% confidence level. Particle size distribution was obtained through an ASTM D792 8-16 hydrometer bulb 152 H standard test. The airflow velocity was measured at Spot 6 (chainage 2 + 200). The velocity data were obtained between 1 September and 30 September 2016, between 1 June and 30 September 2017, as well as between 8 February, and 28 February 2018. Met Set configuration software was applied to a wind speed detector (Met Pak Pro by Rational

Technologies Private Limited, Sonipat, India), with +2% accuracy at 12 m/s and 0.01 m/s resolution. The data were collected every 30 min for 24 h every day.

## 3. Results and Discussion

### 3.1. Construction Stage

The concentrations of gases detected in the Lowari Tunnel during the construction stage are shown in Table 1. Minimum, maximum, and average values of each monitored day are presented for all six gases. The $H_2S$ content changed slightly between 0 and 2.8 ppm, far below the threshold limit value. NO levels varied between 0 and 26.3 ppm, CO levels ranged between 0 and 450 ppm, while $CO_2$ levels ranged between 0 and 14,333 ppm. NO surpassed the low alarming threshold (22.5 ppm) on three days and the high alarming threshold (25 ppm) on one day during the reported period. CO levels exceeded the low alarming threshold of 35 ppm for 23 days and the high alarming threshold of 200 ppm for 8 days. $CO_2$ levels surpassed the low and high warning levels for 6 and 3 days, respectively. The maximum concentrations of $H_2S$, NO, CO, and $CO_2$ (i.e., 2.8 ppm, 26.3 ppm, 450 ppm, and 14,333 ppm, respectively) were detected on 6 December 2008 at 16:48 for Chainage 4 + 549 when the ventilation system was turned down following an explosion due to generator fuel shortage. Finally, oxygen was found to be acceptable (>19.5% minimum) and its concentration did not vary considerably throughout the recorded duration.

Figure 2 depicts the hourly concentration trends of gases in the Lowari Tunnel during blasting and mucking operations. Figure 2a demonstrates that 15 min after the explosion, the NO concentration reached a maximum of 26.3 ppm. During the subsequent 45 min (mucking activity), NO concentrations either declined gradually or stayed constant ($\approx$3 ppm). Figure 2b shows a significant increase in CO concentrations after the explosion, which peaked at 450 ppm in 12 min. The next 30 min (mucking activity) were marked by a considerable decrease in concentration, which was followed by 20 min of stable concentration. Figure 2c depicts a minor rise in the $CO_2$ concentration around 5 min after the explosion; nevertheless, the concentration observed during mucking activity on each subsequent day remained constant. A significant variation (about 30-fold) may be detected between the lowest and highest concentrations (450 ppm and 14,000 ppm, respectively). During the gas monitoring period of the construction stage of the Lowari Tunnel, NO, CO, and $CO_2$ levels exceeded permissible limits. The explosives used for blasting, as well as the loaders and dumpers used for mucking, caused excessive emissions. During the blasting action, NO exhibited the maximum increase, followed by CO and $CO_2$. According to the medical record register of the tunnel doctor, the high concentrations of these gases caused chest and throat diseases among the laborers.

**Table 1.** Gas concentrations measured during the construction stage of Lowari Tunnel at selected spots.

| Month | Date | $H_2S$ (ppm) | | $SO_2$ (ppm) | | NO (ppm) | | CO (ppm) | | $CO_2$ (ppm) | | Oxy (%) | |
|---|---|---|---|---|---|---|---|---|---|---|---|---|---|
| | | Range | Ave. | Range | Ave. | Range | Ave. | Range | Ave. | Range | Ave. | Range. | Ave. |
| November 2008 | 1 November 2008 | 0.0 to 0.3 | 0.1 | 0.0 to 0.0 | 0.0 | 1.2 to 5.5 | 3.3 | - | - | - | - | - | - |
| | 3 November 2008 | - | - | - | - | - | - | 0.3 to 126.5 | 49.2 | 367.0 to 1670.0 | 771.9 | 30.0 to 30.0 | 30.0 |
| | 4 November 2008 | 0.0 to 0.3 | 0.1 | 0.0 to 0.0 | 0.0 | 1.1 to 11.7 | 5.7 | 3.4 to 215.4 | 77.2 | 616.0 to 915.0 | 772.9 | 30.0 to 30.0 | 30.0 |
| | 5 November 2008 | 0.0 to 0.3 | 0.0 | 0.0 to 0.0 | 0.0 | 0.0 to 16.7 | 5.0 | 0.0 to 446.2 | 103.8 | 185.0 to 2151.0 | 611.2 | 30.0 to 30.0 | 30.0 |
| | 6 November 2008 | 0.0 to 0.3 | 0.0 | 0.0 to 0.0 | 0.0 | 0.3 to 18.0 | 6.2 | 0.0 to 327.0 | 79.0 | 111.0 to 2182.0 | 662.8 | 30.0 to 30.0 | 30.0 |
| | 9 November 2008 | 0.0 to 0.3 | 0.1 | 0.0 to 0.0 | 0.0 | 0.3 to 4.2 | 3.3 | 0.0 to 22.4 | 15.3 | 116.0 to 1880.0 | 785.6 | 30.0 to 30.0 | 30.0 |
| | 10 November 2008 | 0.0 to 0.2 | 0.1 | 0.0 to 0.0 | 0.0 | 1.8 to 5.1 | 3.4 | 21.7 to 28.8 | 24.5 | 617.0 to 816.0 | 687.3 | | |
| | 11 November 2008 | 0.0 to 0.1 | 0.0 | 0.0 to 0.0 | 0.0 | 1.8 to 3.1 | 2.5 | 12.1 to 50.4 | 26.2 | 352.0 to 693.0 | 526.4 | 30.0 to 30.0 | 30.0 |
| | 12 November 2008 | 0.0 to 0.3 | 0.0 | 0.0 to 0.0 | 0.0 | 0.7 to 6.1 | 2.6 | 6.0 to 59.7 | 22.9 | 417.0 to 1237.0 | 669.5 | 30.0 to 30.0 | 30.0 |
| | 14 November 2008 | 0.0 to 1.5 | 0.2 | 0.0 to 0.0 | 0.0 | 0.2 to 20.4 | 6.6 | 0.0 to 367.0 | 95.6 | 1559.0 to 3240.0 | 2103.9 | 29.6 to 30.0 | 29.9 |
| | 15 November 2008 | - | - | - | - | - | - | 0.0 to 77.6 | 45.6 | 1192.0 to 2942.0 | 1598.7 | 29.6 to 30.0 | 29.9 |
| | 16 November 2008 | 0.0 to 0.5 | 0.1 | 0.0 to 0.0 | 0.0 | 0.0 to 6.2 | 2.6 | 0.2 to 164.4 | 42.7 | 380.0 to 1641.0 | 759.2 | 30.0 to 30.0 | 30.0 |
| | 17 November 2008 | 0.0 to 0.2 | 0.0 | 0.0 to 0.0 | 0.0 | 0.5 to 4.8 | 2.2 | 1.4 to 32.6 | 11.6 | 397.0 to 2453.0 | 1431.4 | 29.7 to 30.0 | 29.9 |
| | 18 November 2008 | 0.0 to 0.2 | 0.0 | 0.0 to 0.0 | 0.0 | 0.1 to 3.5 | 2.2 | 0.9 to 69.4 | 29.6 | 487.0 to 2121.0 | 972.4 | 29.7 to 30.0 | 29.9 |
| | 19 November 2008 | - | - | - | - | - | - | 0.5 to 403.0 | 92.2 | 434.0 to 2463.0 | 831.6 | 29.8 to 30.0 | 30.0 |
| | 28 November 2008 | - | - | - | - | - | - | 0.0 to 94.8 | 30.3 | 1295.0 to 2692.0 | 2077.6 | 30.0 to 30.0 | 30.0 |
| | 30 November 2008 | 0.0 to 0.0 | 0.0 | 0.0 to 0.0 | 0.0 | 0.4 to 3.9 | 2.6 | 0.0 to 19.5 | 13.2 | 2836.0 to 5888.0 | 5275.1 | 29.8 to 30.0 | 29.9 |
| December 2008 | 1 December 2008 | 0.0 to 0.0 | 0.0 | 0.0 to 0.0 | 0.0 | 0.0 to 3.2 | 2.0 | 9.3 to 48.0 | 30.8 | 2935.0 to 11,777.0 | 7911.5 | 29.3 to 30.0 | 29.4 |
| | 2 December 2008 | 0.0 to 0.0 | 0.0 | 0.0 to 0.0 | 0.0 | 1.4 to 4.3 | 3.2 | 4.6 to 33.6 | 14.5 | 183.0 to 1946.0 | 879.6 | 29.0 to 30.0 | 29.9 |
| | 4 December 2008 | 0.0 to 0.2 | 0.0 | 0.0 to 0.0 | 0.0 | 0.6 to 4.1 | 3.4 | 3.9 to 28.8 | 22.4 | 1402.0 to 3499.0 | 2070.2 | 30.0 to 30.0 | 30.0 |
| | 5 December 2008 | 0.0 to 0.2 | 0.0 | 0.0 to 0.0 | 0.0 | 0.7 to 3.8 | 3.3 | 13.5 to 31.4 | 21.0 | 647.0 to 1692.0 | 860.3 | 30.0 to 30.0 | 30.0 |
| | 6 December 2008 | 0.0 to 2.8 | 0.2 | 0.0 to 0.0 | 0.0 | 1.3 to 26.3 | 7.7 | 4.8 to 450.0 | 109.6 | 534.0 to 1109.0 | 667.7 | 30.0 to 30.0 | 30.0 |
| | 25 December 2008 | 0.0 to 1.9 | 1.0 | 0.0 to 0.0 | 0.0 | 0.0 to 9.5 | 5.6 | 0.0 to 203.9 | 98.2 | 1319.0 to 4934.0 | 3941.4 | 29.7 to 30.0 | 29.9 |
| | 26 December 2008 | 0.0 to 0.2 | 0.0 | 0.0 to 0.0 | 0.0 | 0.0 to 3.0 | 1.4 | 0.1 to 47.4 | 6.6 | 0.1 to 47.4 | 6.7 | 29.6 to 30.0 | 29.9 |
| | 27 December 2008 | 0.0 to 0.3 | 0.1 | 0.0 to 0.0 | 0.0 | 0.0 to 3.9 | 1.8 | 0.0 to 100.4 | 45.3 | 273.0 1072.0 | 630.7 | 29.9 to 30.0 | 30.0 |
| | 28 December 2008 | 0.0 to 0.0 | 0.0 | 0.0 to 0.0 | 0.0 | 0.4 to 1.3 | 0.6 | - | - | - | - | - | - |
| | 29 December 2008 | 0.0 to 0.0 | 0.0 | 0.0 to 0.0 | 0.0 | 0.0 to 2.7 | 1.4 | 0.0 to 10.1 | 3.9 | 659.0 to 2454.0 | 1875.5 | 29.6 to 30.0 | 29.9 |
| | 31 December 2008 | 0.0 to 0.2 | 0.0 | 0.0 to 0.0 | 0.0 | 1.7 to 2.8 | 2.3 | 41.3 to 95.0 | 75.1 | 3040.0 to 7389.0 | 4961.4 | 29.4 to 30.0 | 29.7 |

**Table 1.** *Cont.*

| Month | Date | H$_2$S (ppm) | | SO$_2$ (ppm) | | NO (ppm) | | CO (ppm) | | CO$_2$ (ppm) | | Oxy (%) | |
|---|---|---|---|---|---|---|---|---|---|---|---|---|---|
| | | Range | Ave. | Range | Ave. | Range | Ave. | Range | Ave. | Range | Ave. | Range. | Ave. |
| January 2009 | 1 January 2009 | 0.0 to 0.2 | 0.1 | 0.0 to 0.0 | 0.0 | 0.0 to 8.1 | 3.7 | 1.8 to 228.3 | 65.6 | 1684.0 to 7632.0 | 5906.0 | 29.6 to 30.0 | 29.9 |
| | 3 January 2009 | 0.0 to 0.0 | 0.0 | 0.0 to 0.0 | 0.0 | 0.1 to 1.7 | 1.4 | 0.6 to 56.6 | 29.9 | 1755.0 to 8690.0 | 5880.0 | 29.5 to 30.0 | 29.8 |
| | 5 January 2009 | - | - | - | - | - | - | 0.7 to 52.7 | 38.4 | 6438.0 to 14,333.0 | 7803.9 | 29.3 to 30.0 | 29.4 |
| | 6 January 2009 | 0.0 to 0.0 | 0.0 | 0.0 to 0.0 | 0.0 | 0.0 to 2.5 | 1.5 | 7.8 to 15.6 | 33.2 | 882.0 to 1771.0 | 1528.3 | 29.7 to 29.9 | 29.8 |
| | 7 January 2009 | 0.0 to 0.1 | 0.0 | 0.0 to 0.0 | 0.0 | 0.2 to 3.2 | 2.5 | 0.0 to 78.2 | 48.4 | 473.0 to 2886.0 | 2220.5 | 29.2 to 30.0 | 29.5 |
| | 9 January 2009 | 0.0 to 0.1 | 0.0 | 0.0 to 0.0 | 0.0 | 0.0 to 3.9 | 1.5 | 2.4 to 149.0 | 30.7 | 1046.0 to 2792.0 | 1668.4 | 29.3 to 29.9 | 29.6 |
| | 20 January 2009 | 0.0 to 0.2 | 0.1 | 0.0 to 0.0 | 0.0 | 0.2 to 2.5 | 1.4 | - | - | - | - | - | - |

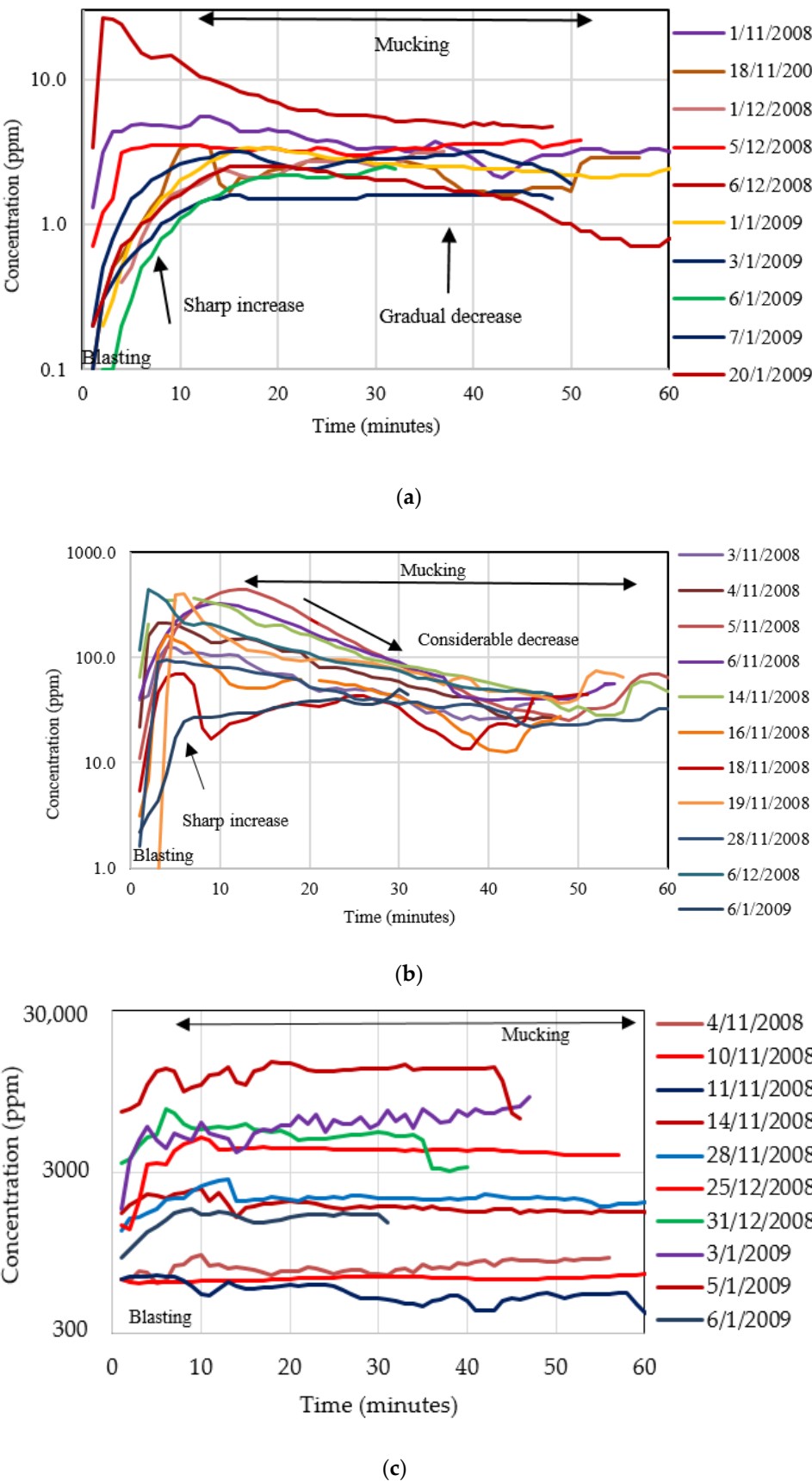

**Figure 2.** Gas concentrations in Lowari Tunnel for 1 h during blasting and mucking activities. (**a**) Concentration of nitrogen oxide; (**b**) concentration of carbon monoxide; (**c**) concentration of carbon dioxide.

### 3.2. Operational Stage

Table 2 displays the average pollution concentrations (gases and particulate matter) recorded on 30 June 2021, during the tunnel's operational phase, at Spots 1 (south portal), 2 (Chainage 1 + 000), 3 (Chainage 4 + 000), and 4 (Chainage 8 + 000). Included among them are NO, CO, $H_2S$, $CO_2$, $O_2$, $O_3$, $NH_3$, $NO_2$, $PM_{10}$, $PM_{2.5,}$ and $PM_1$. Several gases were not detected, including NO, $H_2S$, $O_3$, and $NH_3$. CO was measured to be 1.94 ppm and 5.80 ppm at Spots 1 and 3, respectively. Whereas $CO_2$ was measured at Spots 1 to 4 to be 459 ppm, 369 ppm, 525 ppm, and 484 ppm, respectively. All four locations had appropriate $O_2$ levels (>19.5 ppm), ranging from 22.0 ppm to 24.0 ppm. $NO_2$ concentrations were measured to be 0.002 ppm at all four locations. The measured amounts of gases were well below the Occupational Safety and Health Administration's alarming thresholds (i.e., 35 ppm for CO, 5000 ppm for $CO_2$, and 0.2 ppm for $NO_2$). In addition to mechanical ventilation, natural ventilation was available after tunnel breakthrough, and hence the concentration of gases was within the acceptable limits.

**Table 2.** Gas and particulate matter concentrations during the operational stage.

| | Spot | Chainage | NO | CO | $H_2S$ | $CO_2$ | $O_2$ | $O_3$ | $NH_3$ | $NO_2$ | $PM_{10}$ | $PM_{2.5}$ | $PM_1$ |
|---|---|---|---|---|---|---|---|---|---|---|---|---|---|
| | | | | | | | (ppm) | | | | | | |
| 30 June 2021 | 1 | S. Portal | - | 1.94 | - | 459 | 24.0 | - | - | 0.002 | 0.010 | 0.049 | 0.104 |
| | 2 | 1 + 000 | - | - | - | 369 | 24.0 | - | - | 0.002 | 0.515 | 0.076 | 0.094 |
| | 3 | 4 + 000 | - | 5.80 | - | 525 | 23.1 | - | - | 0.002 | 0.298 | 0.054 | 0.342 |
| | 4 | 8 + 000 | - | - | - | 484 | 22.0 | - | - | 0.002 | 0.371 | 0.208 | 0.061 |
| | Limit | | 22.5 max | 35 max | 10 max | 5000 max | 19.5 min | 0.1 max | 50 max | 0.2 max | 0.054 max | 0.06 max | |

"- Not detected."

Spot 1 at Chainage 1 + 000 near the south portal showed a high concentration of $PM_{10}$ (0.512 ppm), whereas Spot 3 at Chainage 8 + 000 near the north portal showed a high concentration of $PM_{2.5}$ (0.208 ppm). The high concentrations of $PM_{10}$ and $PM_{2.5}$ in the vicinity of the portals are caused by the suspension of road dust. Due to the piston effect from both sides, $PM_{1,}$ with a size of 1 μm, can travel with the vehicle and was concentrated in the middle of the tunnel. At Spots 2, 3, and 4, the $PM_{10}$ levels surpassed the maximum limit (0.054 ppm) (0.515 ppm, 0.298 ppm, and 0.371 ppm, respectively). Similarly, $PM_{2.5}$ levels were above the maximum limit (0.06 ppm) for Spots 2, 3, and 4 (0.076 ppm, 0.054 ppm, and 0.208 ppm, respectively). In contrast, $PM_1$ levels were 0.104 ppm at Spot 1, 0.094 ppm at Spot 2, 0.342 ppm at Spot 3, and 0.061 ppm at Spot 4. However, the maximum allowable $PM_1$ concentration is not established in any standard. Studies have ascribed the source of particulate matter to road traffic [42–47]: $PM_{10}$ and $PM_{2.5}$ is ascribed to non-exhaust emissions including wear and tear (brakes, tires, and clutches), re-suspension of road surface dust, and abrasion [47,48], while $PM_1$ is ascribed to vehicle exhaust [49,50].

Figure 3 depicts the morphological and elemental analyses of the dust samples collected at Spots 1 to 5. In general, angular and subangular particles dominated the micro-morphologies of the dust samples. A few spherical particles were also seen at Spot 1. The dust samples collected at Spots 1 and 5 showed more angular particles and a broad range of particle sizes (the south and north portals, respectively). Overall, the particle sizes were less than 200 μm. Attached to the surfaces of larger particles were smaller particles with a size of 1 μm. The EDS examination revealed that the dust particles were mostly composed of O, C, Si, Fe, Al, Ca, Mg, Zn, Pb, and Pd. The chemical composition of particles inside the tunnel was identical to that of mineral dust at all five spots, caused by exhaust and non-exhaust vehicle emissions [17,51,52].

Figure 4 demonstrates that the composition of road dust at all five spots includes the major elements Si > Ca > Na > Mg > Fe > K and the trace elements Zn > Pb > Ni > Ag > Co > Cd. The chemical composition relates to the mineralogy and traffic of airborne dust (brake and tire wear). These results are comparable with those of Lin [44] and Candeias et al. [53],

who reported the presence of these components in PM$_{1-10}$. The dust particles in the tunnel are supplemented by particles eroded from the concrete lining and road abrasion. The suspended and re-suspended dust particles disrupt tunnel vision and pose a health risk to trail users.

Figure 5 depicts the particle size distribution of the dust samples obtained at Spots 1–5 in the Lowari Tunnel. The total particle size spans from 0.9 to 4750 µm, with Spots 1–5 averaging 170, 130, 130, 80, and 90 µm, respectively. Road dust particle size ranges from 1 to 100 µm according to the World Health Organization [54] and were identified as 30% of the sample obtained at Spot 1, 45% at Spot 2, 45% at Spot 3, and 56% at Spot 4 and Spot 5. These results are congruent with those obtained from the PM concentration, which demonstrated that the PM concentration within the tunnel is greater than that outside. Road dust is the principal source of particulate matter, which is generated by non-exhaust emissions such as wear and tear (brakes, tires, and clutches), re-suspension, road surface dust and abrasion, and exhaust emissions [9,55,56].

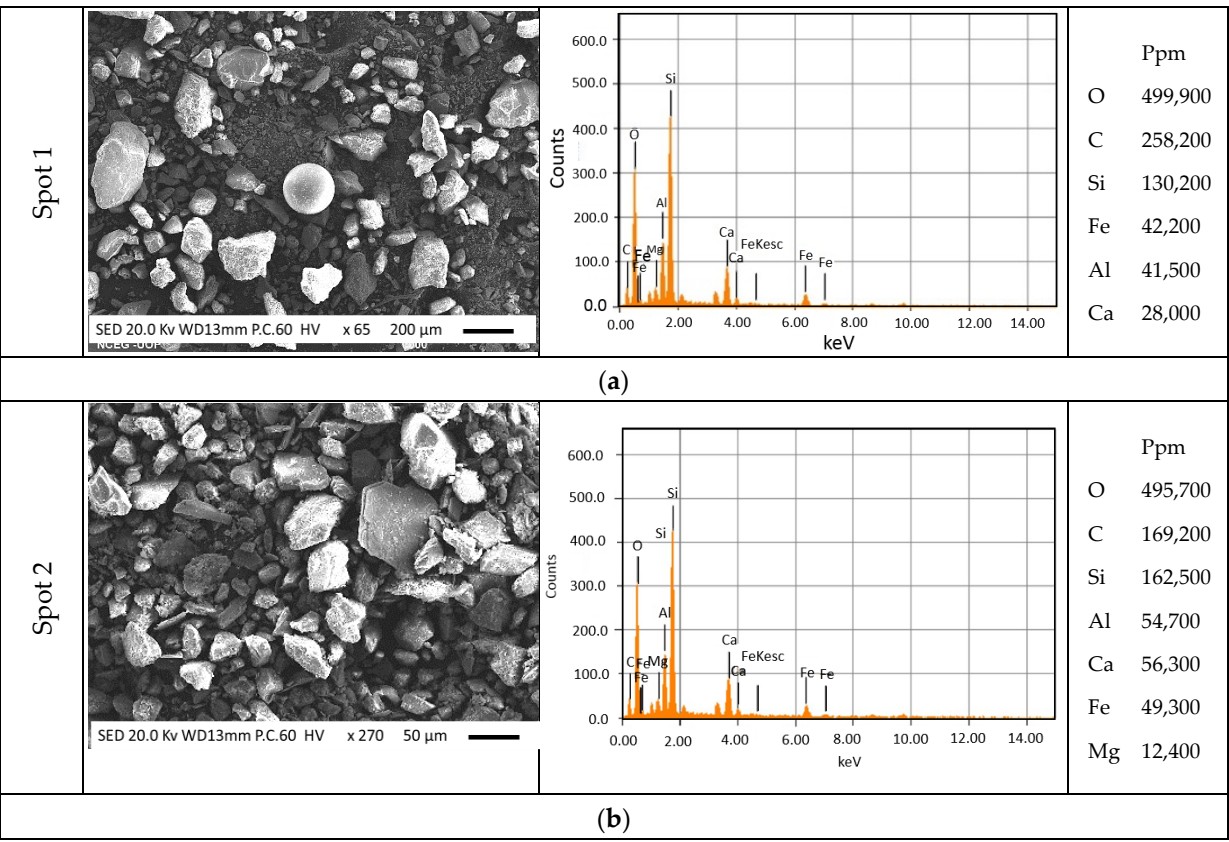

**Figure 3.** *Cont.*

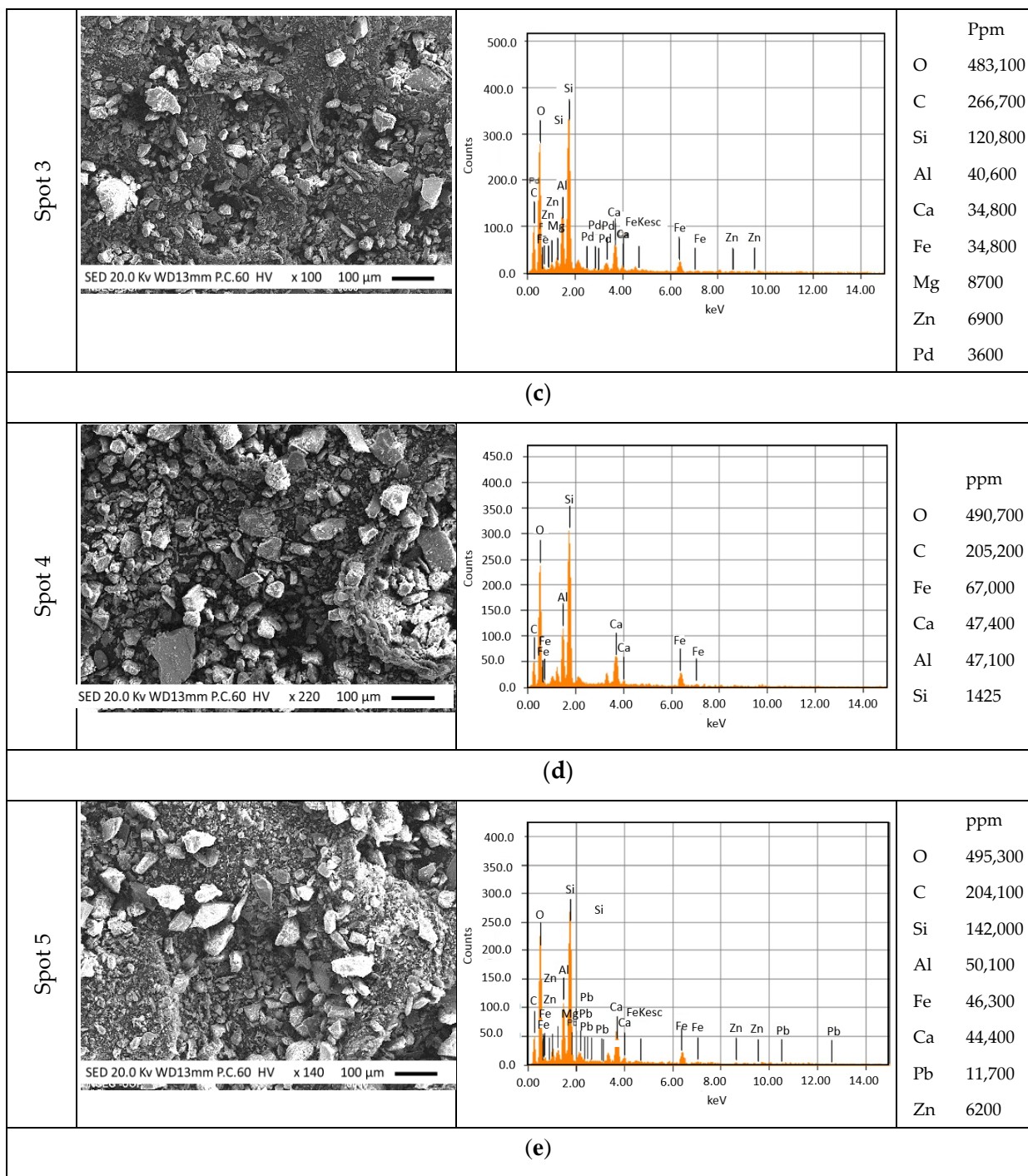

**Figure 3.** SEM and EDS images of Lowari tunnel dust: (**a**) Spot 1; (**b**) Spot 2; (**c**) Spot 3; (**d**) Spot 4; (**e**) Spot 5.

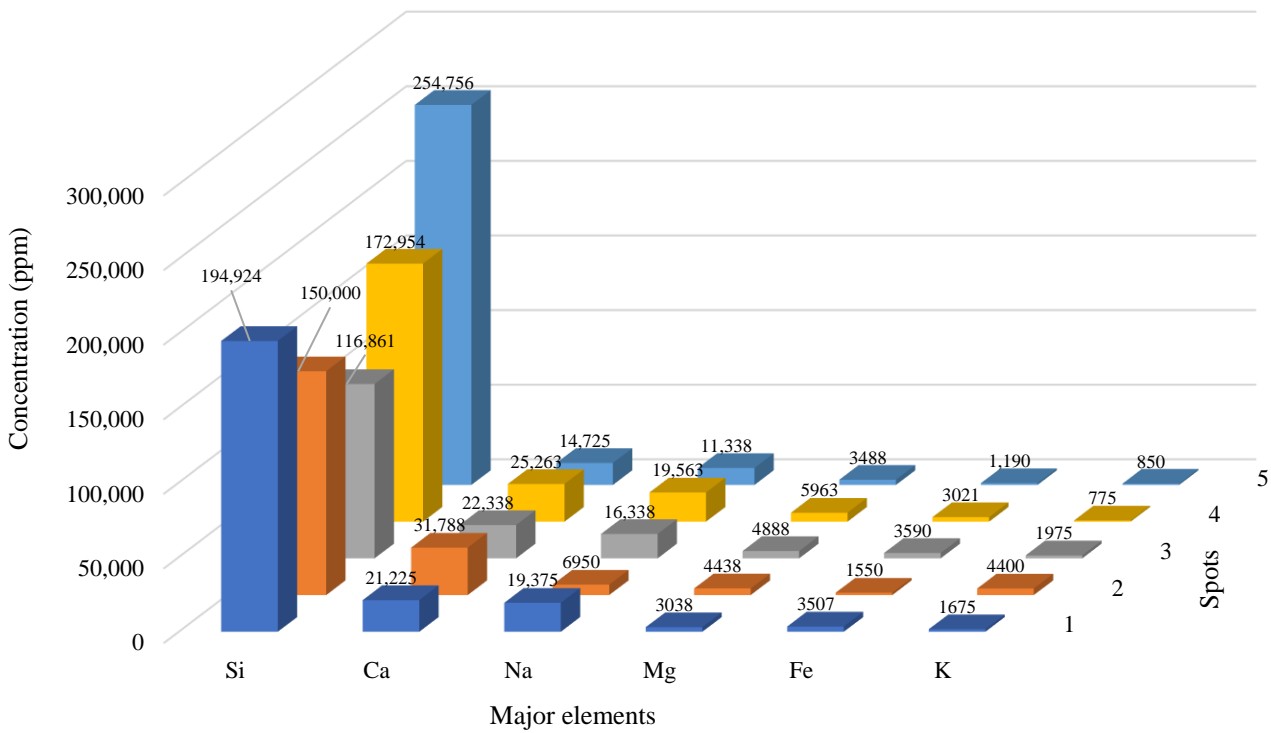

(**a**)

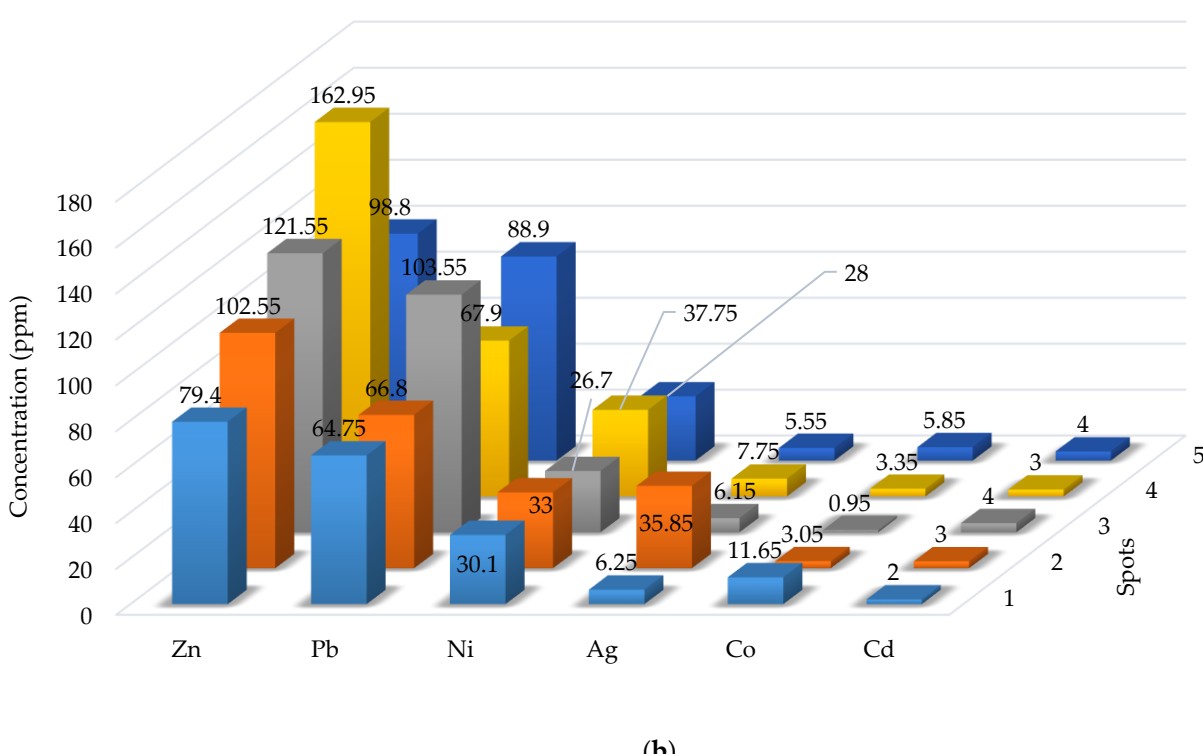

(**b**)

**Figure 4.** Concentrations of major and trace elements of Lowari Tunnel dust. (**a**) Major elements; (**b**) trace elements.

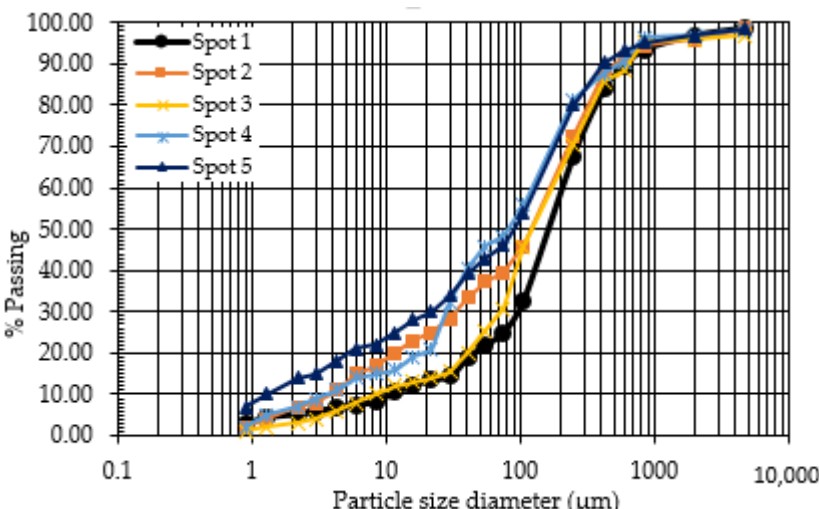

**Figure 5.** Particle size distribution of Lowari Tunnel dust collected at selected spots.

In September 2016, June 2017, and February 2018, the airflow velocity in Lowari Tunnel was measured at Spot 6 (Chainage 2 + 200) during the operational stage. The bars reflect the duration of the airflow, whilst the scatter plot with lines displays the average airflow velocity recorded for 24 h every day, with the standard deviation shown as error bars. The south-to-north airflow velocity is assigned a negative direction, whereas the north-to-south airflow velocity is given a positive direction. For the majority of February 2018 (about 95%), as seen in Figure 6a, the airflow direction was from north to south (ranging from 0.6 m/s to 5.1 m/s). The greater standard deviations in the north-to-south direction show airflow variations, while the smaller standard deviations in the south-to-north direction suggest a stable flow (0 to 3.9 m/s). In June 2017, as seen in Figure 6b, the airflow was mostly north to south (>97%), with a reasonably constant airflow velocity (a range from 1 m/s to 3.7 m/s). In September 2016, as seen in Figure 6c, the direction of the airflow was reversed from south to north throughout the bulk of the month (about 80%), with a constant airflow velocity (a range from 0.8 m/s to 1.5 m/s). The measured statistics indicate that the airflow velocity in Lowari Tunnel fulfills the Safety and Administration [12] minimum airflow velocity for tunnels, bores, shafts, and other subterranean workplaces (0.15 m/s). During the operating phase of the Lowari Tunnel's airflow velocity monitoring period, the amplitude, direction, and fluctuations of airflow were mostly from north to south, with a high airflow velocity in the winter. This is owing to the existence of a high-pressure center in Chitral, north of the Lowari Tunnel, during the majority of the year, particularly in the winter, which causes the airflow direction to change from north to south [57].

Table 3 compares the concentration of pollutants in the Lowari Tunnel with those in the Hsuehshan Tunnel in Taiwan [58], the Xihan ESA 1 Tunnel in China [4], and the Calle Hidalgo Tunnel, the Ponciano Aguilar Tunnel, the Tiburcio Tunnel, the Barretero Tunnel, the Galerena Tunnel, the Los Angeles Tunnel, the Santa F Tunnel, and the Tamazuca Tunnel in Mexico [37]. The CO, $PM_{10}$, and $PM_{2.5}$ concentrations in the Lowari Tunnel are higher than those in other tunnels, most likely as a result of the tunnel's length and the condition of the vehicles traveling through it [37,59,60].

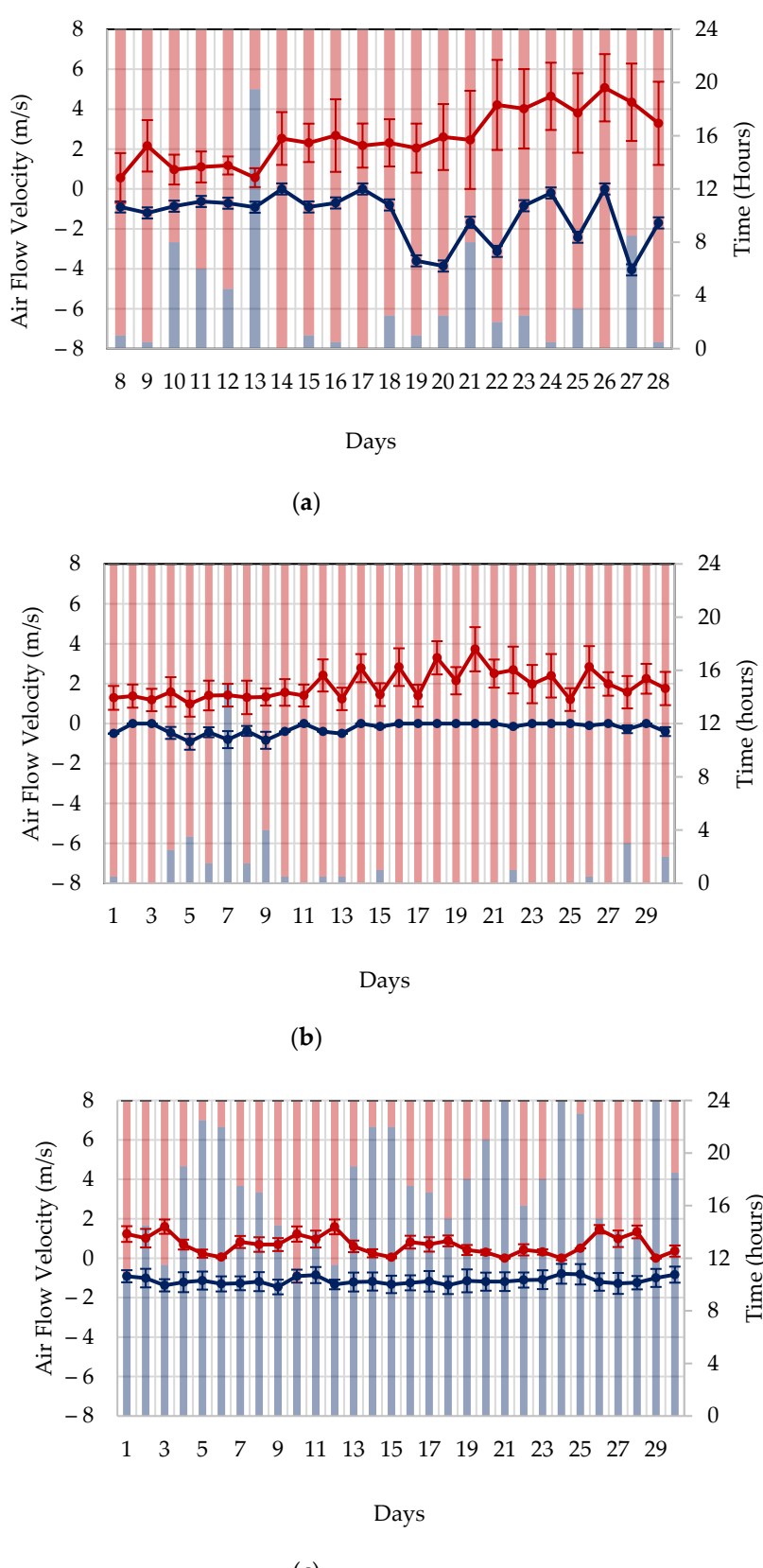

**Figure 6.** Air flow velocity in Lowari Tunnel during the operational stage: (**a**) February 2018; (**b**) June 2017; (**c**) September 2016. The red color shows the wind direction from south to north and the blue color from north to south.

**Table 3.** Comparison of the tunnels' pollutant concentrations during the operational stage.

| Study | Tunnel | Length (m) | Slope % | CO (ppm) | PM$_{10}$ (ppm) | PM$_{2.5}$ (ppm) |
|---|---|---|---|---|---|---|
| This study | Lowari Tunnel | 8509 | 1.8 | 5.80 | 0.40 | 0.11 |
| Li et al., 2011 | Hsuehshan Tunnel | 12,900 | 1.3 | 5.21 | - | - |
| Qian et al., 2019 | Xihan ESA1 Tunnel | 6102 | - | 5.10 | - | - |
| Zamorategui-Molina et al., 2021 | Calle Hidalgo Tunnel | 1100 | 2.1 | 0.01 | 0.08 | 0.08 |
| | Ponciano Aguilar Tunnel | 1100 | - | 0.02 | - | - |
| | Tiburcio Tunnel | 0950 | 1.7 | 0.02 | - | – |
| | Barretero Tunnel | 0800 | 2.6 | 0.03 | 0.10 | 0.09 |
| | Galerena Tunnel | 0700 | 6.7 | 0.02 | 0.08 | 0.08 |
| | Los angeles Tunnel | 0600 | 6.9 | 0.07 | 0.18 | 0.16 |
| | Santa F Tunnel | 0500 | 1.6 | 0.03 | 0.08 | 0.08 |
| | Tamazuca Tunnel | 0110 | 2.1 | 0.02 | - | – |

## 4. Summary and Conclusions

Mostly, tunnel environment research has been undertaken in industrialized countries. To adapt to worldwide environmental standards, building quality, and vehicle conditions, it is necessary to undertake such research in developing nations such as Pakistan. During both the building and operating phases of a tunnel, the monitoring of pollutants such as hazardous gases and particulate matter is essential for maintaining a clean atmosphere. Long tunnels with large slopes, zigzag alignments, water penetration, poor vehicle maintenance, and an insufficient ventilation system pose a high danger of pollutant concentrations exceeding the permitted limits.

During the construction stage (about 30 days), the concentration of gases in the Lowari Tunnel was measured between Chainage 4 + 369.8 and 4 + 595. During the monitored period, NO exceeded the high alarming levels for one day, CO for eight days, and $CO_2$ for three days. $SO_2$ was not detected, and $H_2S$ levels were within the permitted range. $O_2$ levels were determined to be satisfactory (>19.5% minimum) and their concentration did not fluctuate significantly. Maximum amounts of $H_2S$, NO, CO, and $CO_2$ (i.e., 2.8 ppm, 26.3 ppm, 450 ppm, and 14,333 ppm, respectively) were measured in Chainage 4 + 549 at 16:48 on 6 December 2008, when the ventilation system was switched off. Trends in the concentration of gases indicate that, throughout the construction stage, the concentration of measured gases increased significantly after blasting. During the operational stage, the Lowari Tunnel was monitored at four locations for gases and particulate matter. NO, $H_2S$, $O_3$, and $NH_3$ were not identified at any of the four spots, whereas $O_2$, CO, and $NO_2$ concentrations were within acceptable limits. The reduced concentrations are the result of a mix of natural and mechanical ventilation after tunnel breakthrough. Road dust suspensions yielded greater concentrations of PM$_{10}$ and PM$_{2.5}$ near the portals (0.515 ppm and 0.208 ppm, respectively). PM$_1$ may move with the vehicle from both sides and concentrate (0.342 ppm) in the middle of the tunnel (Spot 3) owing to the piston effect.

At five spots between the north and south portals, angular to sub-angular tunnel dust with a broad range of particle sizes was observed. Overall, the particle sizes were less than 200 μm, and smaller particles with diameters of less than 1 μm were attached to the larger particles. The chemical composition of dust included the components Si > Ca > Na > Mg > Fe > K and the trace elements Zn > Pb > Ni > Ag > Co > Cd, indicating the dust's origin as vehicle emissions (both exhaust and non-exhaust). The dust particles in the tunnel are enriched with particles of corroded concrete lining and road abrasion. These suspended and re-suspended dust particles impede tunnel vision and pose health dangers to trail users. The average particle size varied between 80 and 170 μm in diameter, while the dust percentage varied between 30% and 56%. This is consistent with the PM concentration, suggesting that the PM concentration within the tunnel is greater than that outside the tunnel. At Spot 6, the airflow velocity was tested and determined to be satisfactory. The

wind direction was mostly from the north to the south, with variable velocity. Lowari Tunnel was compared with other tunnels across the globe, and it was discovered that its length and vehicle conditions have resulted in a greater concentration of CO, $PM_{10}$, and $PM_{2.5}$ than in other tunnels.

## 5. Recommendations

It is recommended that a proper ventilation system and a pollutant monitoring system be installed. Furthermore, water spray for dust settlement inside the tunnel; washing of the tunnel walls, crown, shoulders, and invert; proper cleaning of the access roads and portals; and avoidance of smoke-producing vehicles are all required.

**Author Contributions:** Conceptualization, J.K. and W.A.; methodology, J.K.; software, J.K. and W.A.; validation, J.K., W.A., H.T.J., G.K. and I.I.; formal analysis, J.K. and M.Y.; investigation, J.K., W.A., I.I., M.Y. and H.T.J.; resources, J.K., W.A. and I.I.; data curation, J.K.; writing—original draft preparation, J.K.; writing—review and editing, H.T.J. and G.K.; visualization, J.K., W.A., M.Y., I.I., H.T.J. and G.K.; supervision, W.A.; project administration, W.A.; funding acquisition, G.K. All authors have read and agreed to the published version of the manuscript.

**Funding:** This research received no external funding.

**Institutional Review Board Statement:** Not applicable.

**Informed Consent Statement:** Not applicable.

**Data Availability Statement:** The data used in this work are available on request.

**Acknowledgments:** The authors would like to acknowledge the National Centre of Excellence in Geology, University of Peshawar, Pakistan, and Pakistan Council of Scientific and Industrial Research (PCSIR), Peshawar, for providing laboratory facilities and field logistics.

**Conflicts of Interest:** The authors declare no conflict of interest.

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
