# Peer review of "Pollutants Concentration during the Construction and Operation Stages of a Long Tunnel: A Case Study of Lowari Tunnel, (Dir–Chitral), Khyber Pakhtunkhwa, Pakistan"

_applsci, doi:10.3390/app12126170_

Round 1
Reviewer 1 Report
The references presented in the literature review are to be described with more details.
More details on the measurement equipment and techniques are to be added.
The collected data is very old, why the paper wasn’t published before?
Details on the data acquisition are to be added. Is it performed manually or using an acquisition system? To be described?
Are the measurements performed hourly, then averaged or only one time per day?
Why was no study performed before the construction stage, as prevention ?
Is there any relation between the weather conditions and the Pollutants Concentration?
Author Response
Reviewer No.1
Replies: Authors are thankful for the review and feedback. Answers to the questions raised are given below.
- The references presented in the literature review are to be described with more details.
Answer: The references in the literature review have been elaborated further. Lines 32-33, 42-43, 44-48, 60-62, 67-69 are examples.
- More details on the measurement equipment and techniques are to be added.
Answer: More information on the measuring equipment and techniques has been provided. Lines 118-172 are relevant.
- The collected data is very old, why the paper wasn’t published before?
Answer: The planning for the Lowari Tunnel began in 1960, and excavation work began in 1975. Due to budgetary and tunnel profile issues, the development progress was intermittent for 42 years. Finally, in 2017, a tunnel with a length of 8.509 kilometers became operational. The data is based on a long period of time. Furthermore, this research article is part of a Ph.D. study for the 2018-2019 academic year.
- Details on the data acquisition are to be added. Is it performed manually or using an acquisition system? To be described?
Answer:
During construction stage:
- Monitoring of gases was conducted during the selected dates of the three months (November 2008, December 2008, and January 2009), using defined instruments, after blasting with a sampling frequency of 60 seconds and an average monitoring duration of one hour in each day. For each day min, max and average values are reported.
During operational stage:
- The gases and PM were monitored for one on 30th June 2021, using defined instrument, for one hour at a one-second frequency and average values are reported.
- For dust analysis, on 30th June 2021 one sample from the tunnel road side of each spot was taken manually and reported. Each sample (about 1000 g) was collected with bristle brush in a tightly sealed plastic bag.
- The wind velocity data were obtained, using defined instrument, in selected days (between September 1 and September 30, 2016, June 1 and September 30, 2017, February 8, and February 28, 2018), taken from Spot 6, at every 30 minutes for 24 hours and average values are reported.
- Are the measurements performed hourly, then averaged or only one time per day?
Answer: The measurement frequency has been addressed in detailed. See answer no- 4.
- Why was no study performed before the construction stage, as prevention?
Answer: The objective of the present study is focused on the pollutant’s concentration during the construction stage and operational stage. Any feasibility studies carried out a few decades ago, before construction, could not reflect the tunnel environment during construction and operation, in addition to the current vehicle conditions, fleet traffic volume, and road conditions. All of these control the pollutants' concentration.
- Is there any relation between the weather conditions and the Pollutants Concentration?
Answer: Weather has an effect on the tunnel pollutants' concentration. For example, rain or snow may interrupt airflow inside the tunnel, thereby increasing the pollutants' concentration. However, studies have shown that during rainy days, lower particle concentrations are observed due to less particle re-entrainment and re-suspension in the tunnel.
Reviewer 2 Report
Dear Authors,
I read with great interest your paper concerning the monitoring of pollutants during the construction and operation phases of the Lowari tunnel. I think that, apart from some minor details (e.g. incomplete or missing units), the manuscript can be accepted with minor revisions.
Best regards
Author Response
Reviewer No. 2
Comments:
Dear Authors,
I read with great interest your paper concerning the monitoring of pollutants during the construction and operation phases of the Lowari tunnel. I think that, apart from some minor details (e.g. incomplete or missing units), the manuscript can be accepted with minor revisions.
Reply: The authors are thankful for the reviews and feedback. The minor details, including incomplete or missing units, have been thoroughly checked and corrected where necessary.
Reviewer 3 Report
The article” Pollutants Concentration during Construction and Operation Stages of a Long Tunnel. A Case Study of Lowari Tunnel, (Dir– Chitral), Khyber Pakhtunkhwa, Pakistan” intends to analyze the air pollutants during the building phase and also during the operational phase studied area: Lowari Tunnel's tunnel Pakistan.
Between November 2008, December 2008, and January 2009, during construction stage were monitored for 30 days H2S, SO2, and NO concentrations and CO, CO2 and O2 concentrations were monitored for 32 days using The IR multi-gas monitor-900614 and the RAE Systems V-Rae PGM-7840 multi-gas detector. The collected readings were then compiled and analyzed using Pro RAE-Suite Software.
During the tunnel's operational phase air pollutants were determined in several locations using the NOVA AMA instrument, a Stack gas analyzer, an Ozone Monitor and HAZ Dust Analyzer, etc.
The article is extremely interesting, clearly explained, and it has a high accuracy in presenting the work phases and research results.
Both sections Materials and Methods and Results and discussion are separated in ”Construction stage” and ”Operational stage” which helps in an easy understanding of results or determinations and analyzes.
Also, the study compares the concentration of pollutants in the Lowari Tunnel to several tunnels from Taiwan, China, USA or Mexico.
I consider that this article may have a major main contribution in the literature and can also be an example for researchers who intends to monitor the construction and use of road tunnel segments and why not roads and highways.
Author Response
Reviewer No. 3
Reviewer comment:
The article “Pollutants Concentration during Construction and Operation Stages of a Long Tunnel. A Case Study of Lowari Tunnel, (Dir– Chitral), Khyber Pakhtunkhwa, Pakistan” intends to analyze the air pollutants during the building phase and also during the operational phase studied area: Lowari Tunnel's tunnel Pakistan.
Between November 2008, December 2008, and January 2009, during construction stage were monitored for 30 days H2S, SO2, and NO concentrations and CO, CO2 and O2 concentrations were monitored for 32 days using The IR multi-gas monitor-900614 and the RAE Systems V-Rae PGM-7840 multi-gas detector. The collected readings were then compiled and analyzed using Pro RAE-Suite Software.
During the tunnel's operational phase air pollutants were determined in several locations using the NOVA AMA instrument, a Stack gas analyzer, an Ozone Monitor and HAZ Dust Analyzer, etc.
The article is extremely interesting, clearly explained, and it has a high accuracy in presenting the work phases and research results.
Both sections Materials and Methods and Results and discussion are separated in” Construction stage” and” Operational stage” which helps in an easy understanding of results or determinations and analyzes.
Also, the study compares the concentration of pollutants in the Lowari Tunnel to several tunnels from Taiwan, China, USA or Mexico.
I consider that this article may have a major main contribution in the literature and can also be an example for researchers who intends to monitor the construction and use of road tunnel segments and why not roads and highways.
Answer: The authors are thankful for the review and feedback. The scope of this research is described in this article on the pollutant’s concentration during the construction stage and operational stage. However, similar methodology can be adopted on roads and highways to monitor the concentration of pollutants, e.g. [1-4].
- Pant, P.; Harrison, R.M. Estimation of the contribution of road traffic emissions to particulate matter concentrations from field measurements: A review. Atmospheric environment 2013, 77, 78-97.
- Klöckner, P.; Seiwert, B.; Weyrauch, S.; Escher, B.I.; Reemtsma, T.; Wagner, S. Comprehensive characterization of tire and road wear particles in highway tunnel road dust by use of size and density fractionation. Chemosphere 2021, 279, 130530.
- Tervahattu, H.; Kupiainen, K.J.; Räisänen, M.; Mäkelä, T.; Hillamo, R. Generation of urban road dust from anti-skid and asphalt concrete aggregates. Journal of hazardous materials 2006, 132, 39-46.
- Gustafsson, M.; Blomqvist, G.; Gudmundsson, A.; Dahl, A.; Swietlicki, E.; Bohgard, M.; Lindbom, J.; Ljungman, A. Properties and toxicological effects of particles from the interaction between tyres, road pavement and winter traction material. Science of the total environment 2008, 393, 226-240.
Round 2
Reviewer 1 Report
After revision the paper can be accepted for publication